# Vital and Nonvital Pulp Therapy in Primary Dentition: An Umbrella Review

**DOI:** 10.3390/jcm11010085

**Published:** 2021-12-24

**Authors:** Luísa Bandeira Lopes, Catarina Calvão, Filipa Salema Vieira, João Albernaz Neves, José João Mendes, Vanessa Machado, João Botelho

**Affiliations:** 1Dental Pediatrics Department, Egas Moniz—Cooperativa de Ensino Superior, 2829-511 Almada, Portugal; catarinacalvao@hotmail.com (C.C.); filipasalemavieira@gmail.com (F.S.V.); 2Clinical Research Unit (CRU), Centro de Investigação Interdisciplinar Egas Moniz (CiiEM), Egas Moniz—Cooperativa de Ensino Superior, 2829-511 Almada, Portugal; jalbernazneves@gmail.com (J.A.N.); jmendes@egasmoniz.edu.pt (J.J.M.); vmachado@egasmoniz.edu.pt (V.M.); jbotelho@egasmoniz.edu.pt (J.B.); 3Endodontics Department, Egas Moniz—Cooperativa de Ensino Superior, 2829-511 Almada, Portugal; 4Evidenced-Based Hub, Centro de Investigação Interdisciplinar Egas Moniz, Egas Moniz—Cooperativa de Ensino Superior, 2829-511 Almada, Portugal

**Keywords:** endodontics, pediatric dentistry, oral health, dental medicine, systematic review, umbrella review

## Abstract

Dental caries is the most common non-communicable disease in children with significant aesthetic, functional, and quality of life deterioration. Depending on the depth, two approaches may be considered in primary dentition: vital pulp therapy (VPT) or non-vital therapy (NPT). This umbrella review aimed to critically assess the available systematic reviews (SRs) on VPT and NPT. An electronic database search was conducted (PubMed, Embase, Scopus, Cochrane, Web of Science, and LILACS) until June 2021. The Risk of Bias (RoB) of SRs was analyzed using the Measurement Tool to Assess SRs criteria 2 (AMSTAR2). From 272 entries, 33 SRs were included. Regarding the methodological quality, three studies were critically low, nine low, seventeen moderate, and six were rated as high quality. The quality of evidence produced by the available SRs was moderate. Future high standard SRs and well-designed clinical trials are warranted to better elucidate the clinical protocols and outcomes of VPT and NPT.

## 1. Introduction

Dental caries is the most common non-communicable disease in children with significant aesthetic, functional and quality of life deterioration [1]. Caries lesions can jeopardize the teeth vitality, as its progression cause infection, pain, and even early tooth loss [1,2]. Thus, a timely intervention is key to avoid unpleasant consequences for the child. Depending on the depth of caries (which may have pulp involvement), two approaches may be considered in the primary dentition: vital pulp therapy (VPT) or non-vital therapy (NPT) [1,3].

When the pulp is still recuperable, VPT may be an option and three options are available: indirect pulp treatment (IPT), direct pulp cap (DPC), and pulpotomy [1,2,3,4,5]. When the caries lesion progresses to the point where pulp necrotizes, then an NPT is performed, such as pulpectomy [3].

The efficacy of VPT and NPT has been widely researched [2,3,4,5]. However, the variability of designs, techniques, and material contributes to high heterogeneity regarding the evidence produced. IPT is a technique that leaves at the bottom of the cavity some deep caries to avoid pulp exposure, being covered with a biocompatible material to produce a biological seal [2,4,5]. DPC is a procedure in which there is a pulp exposure, being covered with a biocompatible material. There is a controversy about this method since it has shown limited success [2,3,4,5]. Pulpotomy is an approach applied when there is a carious pulp exposure and where the entire coronal pulp is removed, hemostasis of the radicular pulp is accomplished, and the remaining radicular pulp is treated with a medicament [3,6]. In contrast, pulpectomy is a nonvital treatment (NVT), being a root canal treatment with irreversibly inflamed or necrotic pulp resulting from caries or trauma [1,3,5,6]. Due to the clinical interest of these procedures in endodontics, several systematic reviews (SRs) have been published. Thus, appraising all the available evidence-based information would be of great interest.

Therefore, this umbrella review aimed to appraise the existing evidence on VPT and NVT in primary teeth. Our main focus was to ascertain the overall clinical efficacy of each procedure and its quality of evidence.

## 2. Materials and Methods

We followed the Preferred Reporting Items for Systematic Reviews and Meta-Analyses (PRISMA) guideline updated in 2020 [7] (Appendix A) and the guide for systematic reviews of systematic review [8]. The review protocol was approved a priori by all authors.

The Review question was: “How effective are VPT and NPT for treating deep carious lesion on primary dentition?”.

The following PECO statements were set: Population (P)—Patients with deep caries on primary dentition; Exposure (E)—Clinical management; Comparison (C)—VPT (IPT, DPC and pulpotomy) and NPT (pulpectomy); Outcome (O)—Diagnosis and a variety of dental treatment types.

### 2.1. Eligibility Criteria

The inclusion criteria were as follows: (1) systematic review (with or without meta-analysis); (2) retrieving data from human studies; (3) addressing VPT and NPT on primary teeth. No restrictions to publication year or language were applied. Grey literature was searched through three appropriate databases (opensigle.inist.fr, https://www.ntis.gov/, https://www.apa.org/pubs/databases/psycextra, all accessed in June 2021).

### 2.2. Information Sources Search

Electronic data search was performed in seven electronic databases: PubMed (via Medline), Scopus, Cochrane Database of Systematic Reviews, Scielo (Scientific Electronic Library Online), EMBASE (The Excerpta Medica Database), LILACS (Latin-American scientific literature in health sciences), and TRIP (Turning Research Into Practise) up to June 2021. We merged keywords and subject headings in accordance with the thesaurus of each database and applied exploded subject headings, with the following syntax “(Primary teeth [MeSH] OR Pulp therapy [MeSH] OR Tooth [MeSH]) AND (Pulpotomy OR Pulpectomy OR Vital pulp therapy OR Deciduous teeth) AND (Systematic Review OR Meta-analysis)”.

### 2.3. Study Selection

Two researchers (FV and CC) independently screened titles and abstracts. The agreement between the reviewers was assessed by Kappa statistics. Any paper classified as potentially eligible by either reviewer was ordered as a full-text and independently screened by the reviewers. All disagreements were resolved through discussion with a third reviewer (LBL).

### 2.4. Data Extraction Process and Data Items

Two researchers (FV and CC) independently extracted: authors and year of publication, objective/focused question, databases searched, number of studies included, type of studies included, main results and main conclusions. All disagreements were resolved through discussion with a third reviewer (LBL).

### 2.5. Risk of Bias Assessment

Two researchers (FV and CC) employed the MeaSurement Tool to Assess Systematic Reviews (AMSTAR 2) to determine the methodological quality of the included reviews [8]. AMSTAR 2 is a comprehensive 16-item tool that rates the overall confidence of the results of the review. According to the AMSTAR guidelines, the quality of the systematic reviews was considered as follows: High means ‘Zero or one non-critical weakness’; Moderate means ‘More than one non-critical weakness’; Low means ‘One critical flaw with or without non-critical weaknesses’; and Critically low means ‘More than one critical flaw with or without non-critical weaknesses. The estimation of the AMSTAR quality rate for each study was calculated through the AMSTAR 2 online tool (https://amstar.ca/Amstar_Checklist.php).

## 3. Results

### 3.1. Study Selection

Electronic searches retrieved a total of 272 titles through the database search. After manual assessment of title/abstract and removal of duplicates, 60 potentially eligible full-texts were screened (Figure 1). Full-text screening excluded thirteen studies with reasons (Appendix A), resulting in thirty-five systematic reviews that fulfilled the inclusion criteria. Inter-examiner reliability at the full-text screening was recorded as high (kappa score = 1.00).

### 3.2. Study Characteristics

In total, 33 systematic reviews [1,2,3,4,5,6,9,10,11,12,13,14,15,16,17,18,19,20,21,22,23,24,25,26,27,28,29,30,31,32,33,34,35] were included in the present umbrella review (Table 1). All SRs covered a defined timeframe; however, one did not mention such information [33]. Three systematics reviews failed to report a language restriction [2,10,11], seventeen restricted their search to studies in English [12,13,14,15,16,17,18,19,20,21,22,23,24,25], one restricted to English and Persia [26], and the remaining had no language restrictions [1,3,4,5,6,9,27,28,29,30,31,32,33,34,35].

### 3.3. Methodological Quality

Regarding the methodological quality of SRs, three studies were assessed as of critically low quality [12,27], nine as of low quality [2,6,15,19,21,25,31,35], seventeen studies as of moderate quality [5,9,10,13,14,18,20,22,23,24,26,28,29,30,32,33,34], and six as of high quality [1,3,4,11,16,17] (detailed in Table 2). None of the included SR fully complied with the AMSTAR2 checklist. Overall, SRs mostly failed on: reporting on the sources of funding for the studies included in the review (93.9%, *n* = 31); providing a satisfactory explanation for, and discussion of, any heterogeneity observed in the results (27.3%, *n* = 9); reporting any potential sources of conflict of interest, including funding sources (27.3%, *n* = 9); explaining their selection literature search strategy (20.0%, *n* = 7).

### 3.4. Synthesis of Results

#### 3.4.1. Vital Pulp Therapy

##### Indirect Pulp Treatment (IPT)

In an IPT approach, the caries lesion is not fully removed during instrumentation to avoid pulp exposure, and the remaining affected dentin is then covered with a biocompatible material as a biological seal [4].

Dentin coverage with a liner provides no benefit to the IPT clinical success either using calcium hydroxide (CH) or inert materials (adhesive system or glass-ionomer cement [GIC] [4,22], and with a certain level of confidence as they are based on SRs of high [4] and moderate methodological quality [22].

Also, IPT demonstrates higher clinical success rate than pulpotomy, with low confidence [6]. The Hall technique (an adapted IPT approach) showed 78% success versus a 76% success of pulpectomy, with moderate confidence [5].

##### Direct Pulp Capping (DPC)

In the DPC approach, the pulp is exposed during caries removal and covered with a biocompatible material [3,4,5].

A high-quality SR concluded that DPC presents an 88.8%, success rate regardless of the applied material (CH, dentin bonding agents, MTA and FC) (Coll 2017). These results are corroborated by a moderate quality SR [18] and a high quality Cochrane SR [3]. MTA or enamel matrix proteins do not present uppermost efficacy than CH, and bonding agent directly upon the exposed pulp without previous etching had no significantly different efficacy when compared to CH, MTA or calcium-enriched cement [34].

Also, DPC was shown to present lower clinical success than pulpotomy with moderate confidence [5].

##### Pulpotomy

A pulpotomy is delivered to exposed pulps during deep caries lesions removal with previously confirmed pulp vitality [4]. Clinically, this approach comprises: total coronal pulp removal; successful hemostasis; and coverage of the remaining radicular pulp with a biocompatible material.

A systematic review and meta-analyses with low quality stated no statistically significant difference in the clinical success rate between pulpotomy and pulpectomies in primary incisors [2].

Regarding the materials required in pulpotomy, the studies were diverse. One systematic review from Cochrane library with high quality stated that the evidence suggests MTA may be the most efficacious medicament to heal the root pulp after pulpotomy [3]. Considering MTA and Biodentine, no significant difference was found in clinical and radiographic success with a moderate quality review [20] and a high-quality review [11]. Already in turn, MTA showed superior long-term treatment outcome than ferric sulfate (FS) with moderate quality review [13], and in three meta-analyses with moderate quality, better rates of clinical and radiographic success than formocresol (FC) [23,26,29] were also mentioned, as well as CH, also with moderate quality review, with good quality of the RCT and homogeneity among the studies [24]. Another meta-analyses with moderate quality that addressed FC and FS demonstrated no significant difference in terms of clinical and radiographic outcomes [10]. Two studies with low quality that compared MTA, Biodentin, FC, and SF agreed that there was no significant difference between those materials, but MTA was considered a better option [15,35], the quality of evidence on the included studies being in one a systematic review regarding the comparisons of Biodentine and formocresol, as well as Biodentine and ferric sulfate, low and very low, respectively [35]. On the other hand, a systematic review with low quality referred MTA as the material of choice, and CH with the worst clinical performance [15]. Another systematic review with low quality compared the previously stated materials with herbal medicines (allium satvum, ankaferd blood stopper, elaegnus angustifolia, propolis), which found similar clinical and radiographic success rates when compared to the usual pulpotomy materials, the overall quality of research in the clinical success of herbal medicine as a pulpotomy medicament not being adequate [25]. One other systematic review with low quality assessed the effectiveness of FS, which reported a high success rate, but the studies included in the review were with limited evidence of high-quality studies [21]. A meta-analysis with moderate quality presented by Lin et al. (2014) showed MTA had the best performance, followed by FC and CH, and CH had more failures than FC and FS [28].

Seven systematic reviews also investigated the effect of lasers as well as the materials mentioned above. Several types of lasers were considered, such as diode, Er:YAG, Nd:YAG, He-Ne, CO2, and low level laser. One meta-analysis with low methodological quality showed no statistically significant difference in clinical and radiographic outcomes between laser pulpotomy and conventional pulpotomy [16]. Another meta-analysis with critically low quality revealed that laser had superior clinical results at a 36-month follow-up period [12]. On the contrary, two studies reported that laser had inferior success than conventional pulpotomy techniques, one being of critically low quality [27], and the other with moderate quality [28]. Finally, another systematic review with low quality showed no significant differences in clinical and radiographic pulpotomy outcomes with laser compared with other techniques [31]. One other meta-analysis with high quality compared different materials, referring that MTA and FC success rates were the highest of all pulpotomy types, including laser, were not significantly different [4]. A recent systematic review from the Cochane Library with high quality stated that the evidence shows MTA may be the best material for pulpotomy of primary teeth. However, other materials should be considered as alternatives like Biodentine, enamel matrix derivative, laser treatment or Ankaferd Blood Stopper. When these materials are not available, application of sodium hypochlorite is the safest option [3].

Note that several studies with different quality, but most with moderate quality mentioned that FC, because of its constitution that presents formaldehyde, presents a concern due its potential carcinogenicity, mutagenicity, and cytotoxicity effect [3,4,10,12,13,15,16,18,21,23,26,28,29,31,34,35].

#### 3.4.2. Non-Vital Pulp Therapy (NPT)

Among the NVT techniques, pulpectomy of primary teeth is indicated when irreversible pulpitis or necrotic pulp occurs [1].

As far as obturation concerns, resorbable materials are mandatory. Zinc oxide eugenol (ZOE) pulpectomies yielded similar outcome than Vitapex and Sealapex, with moderate methodological confidence [14]. A systematic review from Cochrane with high quality mentioned no conclusive evidence that one medicament or technique is superior to another. Therefore, the choice of medicament remains at the clinician’s discretion, since comparison between Metapex and (ZOE) paste was inconclusive, as well as Endoflas and ZOE, and finally suggested ZOE paste may be better than Vitapex [3]. In pulpectomy on primary teeth nearing exfoliation, Ca(OH)_2_/iodoform the best filling material to be used for pulpectomy in primary teeth nearing exfoliation, being moderate risk [30]. One other study considered zinc oxide eugenol/iodoform/calcium hydroxide or ZOE fillers perform better than iodoform filler, where the risk of bias was high despite heterogeneity between the studies [1]. Regarding lesion sterilization and tissue repair (LSTR) technique and pulpectomy, two meta-analyses showed a nonsignificant difference, one being with moderate quality [9], and another one with high quality [1].

Concerning smear layer, there is no consensus, with one study considering studies at a moderate risk of bias [5,32], despite one systematic review only having taken into account two studies [32]. The root canal irrigation has several products despite the controversy on their performance, as the study exhibited moderate risk of bias [33].

Regarding rotary instrumentation, there are similar clinical and radiographic success rate, but with a better-quality treatment in less time [1,19], despite the evidence showing low quality [19] and high quality, although there was heterogeneity among the studies [1]. On the contrary, another study referred there were not enough studies to assess whether rotary versus manual instrumentation affects clinical and radiological success, with a publication bias low [17].

## 4. Discussion

This umbrella review clearly summarizes the evidence sourced in VPT and NPT in primary teeth. The methodological quality of the included SRs ranged from very low to high quality, and therefore current evidence is of moderate confidence.

A correct diagnosis of the pulp status is crucial for correct treatment options and therefore for prognosis. In this regard and attending that the preservation of pulp vitality is fundamental, VPT treatment approaches must be considered. IPT allows selective tissue removal when compared to DPC and pulpotomy. IPT does not expose or damage the pulp and allows it to recover and heal by itself; this way selective caries removal are recommended [4,5,6,22]. However, the hall technique should be considered since it showed a superior success rate compared to non-selective and selective caries removal [5]. On the contrary, DPC is controversial, since there is not sufficient support to recommend it [4,5,33,34,36], due to its great occurrence of complications like mobility, percussion sensitivity, swelling, parulis, or presence of fistulous tract [5]. It is considered that DPC can succeed in case of vital pulp or reversible pulpitis without evidence of radicular pathology [5,36] and appropriate sealing of the cavity [5,34,36]. Already, although considering the pulpotomy is an acceptable and common procedure in case of deep caries, its success depends on several factors such as removing caries prior pulp exposure to avoid pulp contamination, rubber dam isolation application of different medicaments, and experience of the professional [4,11].

Regarding medicaments, the studies sometimes compare only two or several medicaments, MTA, Biodentine, FC, SF, and sodium hypochlorite being considered suitable despite the heterogeneity of the studies [2,3,4,10,11,13,15,20,21,23,26,29,35]; however, there is an agreement that CH is considered to have a less success rate [4,15,23]. In this sense, more well-designed studies with longer follow-up periods and superior methodology are required in order to obtain high evidence [2,4,10,11,15,29,35]. One systematic review highlighted the clinical and radiographic success rates of herbal medicine being suitable replacements to standard pulpotomy medicaments, but due to the heterogeneity of the studies and commercial availability, more studies are necessary to achieve alternatives to the standard medication [25]. Another option has been considered—the laser pulpotomy—which has been described as controversial in its results. Several factors must be determined such as pulp diagnosis, longer observation times, control group, and evaluation of different types of lasers [12,16,27,28,31]. It is also necessary to take into account that there is a learning curve for laser application [27] and there are some advantages of the laser on children like less chair time, painless treatment, and no high-speed rotors [16]; thus, more well-designed randomized clinical trials are required.

The lifecycle of primary teeth are fundamental to a normal growth and development of arch length and occlusal balance [33]. Therefore, sometimes, pulpectomy is necessary to keep the tooth in the arch. However, this procedure is a challenge because of the characteristics of the root canal system like side channels and accessories at the apex and furcation regions, as well as the root anatomy itself and the proximity of the apex to the germs of the permanent tooth [9,32]. Thus, it is of the utmost importance to consider several aspects of the clinical procedure, such as initial pulp condition, type of teeth, manual versus mechanical instrumentation, irrigants used, number of visits, root canal filling material, and type of restoration [14,32]. Therefore, further studies are important with a bigger sample, higher methodological quality, and particularly with longer follow-up, given the controversy between the studies.

### Strengths and Limitations

The present umbrella review benefits from its comprehensive review of the available SRs using a transparent methodology. However, one limitation does need to be accounted for when interpreting the results. In each SR, the individual studies included were not explored. Thus, the conclusions of this review are based on the interpretation of the authors.

## 5. Conclusions

Both VPT and NPT present high clinical efficacy in primary teeth. The results should be interpreted with caution, as the quality of SRs included is overall moderate. Well-designed clinical trials and high standard systematic reviews are necessary to verify the efficacy of treatment options, clinical outcome efficacy, and material suitability.

## Figures and Tables

**Figure 1 jcm-11-00085-f001:**
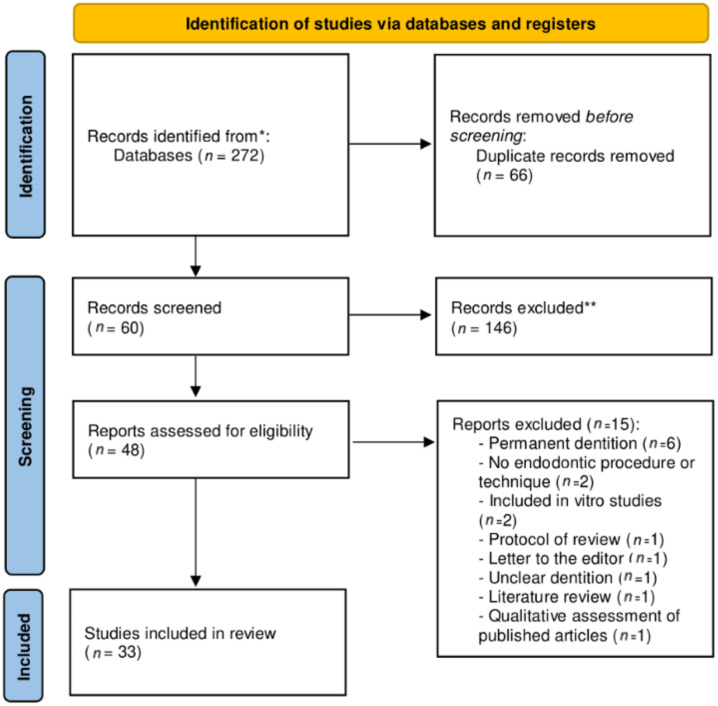
PRISMA flowchart of included studies.

**Table 1 jcm-11-00085-t001:** Characteristics of included studies.

Authors (Year)	N	Search Period	Interventions	Quality Assessment Tool	Sample	Method of Analysis	Outcomes	AMSTAR2 Score *	Funding
Ansari et al. (2018) [12]	17	Up to November 2017	Laser vs. FC in pulpotomy	None	15 NRSI and 2 case reports	SR & MA	Success rate (clinical and radiographic)	Critically Low	NI
Asgary et al. (2014) [13]	4	Up to June 2013	MTA vs. FS in pulpotomy	Modified van Tulder list [1]	4 RCTs	SR & MA	Success rate (clinical)	Moderate	NI
Barcelos et al. (2011) [14]	2	Up to May 2017	ZOE vs. No ZOE pulpectomy	Jadad’s scale [2]	2 RCTs	SR	Success rate (clinical and radiographic)	Moderate	NI
Bossu et al. (2020) [15]	41	Up to October 2019	MTA vs. Biodentine vs. FS vs. FC in pulpotomy	Cochrane Collaboration Tool	NI	SR	Success rate (clinical and radiographic)	Low	Self-funded
Chandran et al. (2020) [16]	14	Unclear (up to 2020)	Laser pulpotomy vs. conventional pulpotomy	Cochrane Collaboration Tool	14 RCTs	SR & MA	Success rate (clinical and radiographic)	High	NI
Chugh et al. (2020) [17]	11	Up to March 2020	Rotary vs. hand root canal instrumentation	ROB 2.0 [3]	11 RCTs	SR & MA	Success rate (clinical)	High	NI
Coll et al. (2017) [4]	87	Since 1990	Indirect Pulp Therapy vs Direct pulp capping Vs Pulpotomy	ROB	-	SR & MA	Success rate (clinical and radiographic)	High	NR
Coll et al. (2020) [1]		Unclear (up to 2020)	Pulpectomy rate success in teeth with and without root resorption,	ROB	-	SR & MA	Success rate (clinical and radiographic)	High	NR
De Coster et al. (2013) [27]	7	Unclear (up to 2012)	Laser vs. conventional pulpotomy procedures	Dutch Cochrane Collaboration tool	5 RCTs and 2 Case series	SR	Success rate (clinical)	Critically Low	NI
Duarte et al. (2020) [9]	6	Up to December 2019	Lesion sterilization and tissue repair (LSTR) pulpotomy vs. pulpectomy	Cochrane Collaboration Tool	6 RCTs	SR & MA	Success rate (clinical and radiographic)	Moderate	Research Grant
Gadallah et al. (2018) [2]		Up to March 2018	Pulpotomy Vs pulpectomy	Cochrane Collaboration Tool	4 RCTs	SR & MA	Success rate (clinical and radiographic)	Low	Self-funded
Garrocho Rangel et al. (2019) [18]	12	Up to December 2019	Direct pulp capping with no carious or small carious exposure of pulpt	Criteria developed by the authors	12 RCTs	SR	Success rate (clinical and radiographic)	Moderate	Partially by Research Grant
Ghajari et al. (2008) [26]	8	Up to March 2008	MTA vs. FC in pulpotomy	Jadad’s scale [2]	8 RCTs	SR & MA	Success rate (clinical and radiographic)	Moderate	NI
Junior et al. (2019) [11]	9	Up to August 2017	Biodentine vs. MTA in pulpotomy	Cochrane Collaboration Tool	7 RCTs and 2 NRSI	SR & MA	Success rate (clinical and radiographic)	High	NI
Lin et al. (2014) [28]	37	Up yo December 2012	MTA vs. Biodentine vs. FS vs. FC vs. Laser in pulpotomy	Criteria developed by the authors	37 RCTs	SR and Network MA	Success rate (clinical and radiographic)	Moderate	Research Grant
Manchanda et al. (2020) [19]	13	Up to January 2019	Rotary vs. hand root canal instrumentation	ROB 2.0 [3]	13 RCTs	SR & MA	Success rate (clinical and radiographic)	Low	NI
Marghalani et al. (2014) [29]	20	Up to May 2013	MTA vs. FC in pulpotomy	Cochrane Collaboration Tool	20 RCTs	SR & MA	Success rate (clinical and radiographic)	Moderate	NI
Nagendrababu et al. (2018) [20]	8	Up to October 2017	MTA vs. Biodentine in pulpotomy	ROB 2.0 [3]	8 RCTs	SR & MA	Success rate (clinical and radiographic)	Moderate	NI
Najjar et al. (2019) [30]	15	Up to January 2018	CH/iodoform vs ZOE in pulpectomy	CONSORT [4]	15 RCTs	SR & MA	Success rate (clinical and radiographic)	Moderate	Self-funded
Nematollahi et al. (2019) [31]	12	Up to September 2017	Laser vs no laser pulpotomy	Jadad’s scale [2]	12 RCTs	SR & MA	Success rate (clinical and radiographic)	Low	Self-funded
Nuvvula et al. (2018) [21]	20	Up to January 2017	FS vs. other agents in pulpotomy	Fuks and Papagiannoulis criteria [5]	NI	SR	Success rate (clinical and radiographic)	Low	Self-funded
Peng et al. (2007) [10]	11	Up to May 2006	FC vs. FS in pulpotomy	Jadad’s scale [2]	4 RCTs, 4 CCTs, 3 retrospective studies	SR & MA	Success rate (clinical and radiographic)	Moderate	NI
Pintor et al. (2016) [32]	2	Up to May 2013	Smear layer removal vs non removal	Cochrane Collaboration Tool	2 RCTs	SR	Success rate (clinical and radiographic)	Moderate	NI
Pozos-Guillen et al. (2016) [33]	7	NI	Clinical efficacy of intracanal irrigants in pulpectomy	Criteria developed by the authors	7 RCTs	SR & MA	Success rate (clinical)	Moderate	Research Grant
da Rosa et al. (2019) [22]	17	Up to February 2018	CH vs. no-CH as pulp capping	Cochrane Collaboration Tool	14 RCTs and 1 retrospective study on primary teeth	SR & MA	Success rate (clinical)	Moderate	Research Grant
Schwendicke et al. (2016) [34]	11	Up to April 2015	Comparing direct pulp capping materials	Cochrane Collaboration Tool	11 RCTs	SR & MA	Success rate (clinical and radiographic)	Moderate	Self-funded
Shafaee et al. (2019) [35]	10	Up to July 2018	MTA vs. Biodentine vs. FS vs. FC in pulpotomy	Cochrane Collaboration Tool	10 RCTs	SR & MA	Success rate (clinical and radiographic)	Low	NI
Shirvani et al. (2014 a) [23]	19	Up to March 2013	MTA vs. FC in pulpotomy	Modified van Tulder list [1]	19 RCTs	SR & MA	Success rate (clinical)	Moderate	Self-funded
Shirvani et al. (2014 b) [24]	4	Up to March 2013	MTA vs. CH in pulpotomy	Modified van Tulder list [1]	4 RCTs	SR & MA	Success rate (clinical and radiographic)	Moderate	NI
Smaïl-Faugeron et al. (2016) [6]		Up to February 2015	Indirect pulp capping Vs Pulpotomy	Cochrane Collaboration Tool	8 Survey of dental prattise, 1 non-randomized study, 2 protocols of ongoing randomized trials	SR	Success rate (clinical and radiographic)	Low	NR
Smaïl-Faugeron et al. (2018) [3]	87	Up to August 2017	MTA vs. Biodentine vs. FS vs. FC vs. Laser in pulpotomy	Cochrane Collaboration Tool	87 RCTs	SR & MA	Success rate (clinical and radiographic)	High	NR
Subramanyam et al. (2017) [25]	8	Up to May 2017	Herbal medicines vs. standard pulpotomy	Criteria developed by the authors	8 RCTs	SR	Success rate (clinical and radiographic)	Low	Self-funded
Tedesco et al. (2021) [5]	9	Up to May 2020	Best approach for deep caries lesion	Cochrane Collaboration Tool	9 RCTs	SR & MA	Success rate (clinical)	Moderate	Self-funded

CCTs—controlled clinical trials; CH—calcium hydroxide; FC—formocresol; FS—Ferric Sulfate; MA—Meta-Analysis; MTA—mineral trioxide aggregate; N—number of included studies; NRSI—Nonrandomized study of intervention; RCTs—randomized-clinical trials; SR—Systematic Review; ZOE—zinc oxide eugenol; NI—no information; NR—not reported. * Detailed information regarding the methodological quality assessment is present in Table 2.

**Table 2 jcm-11-00085-t002:** Results of the methodological quality assessment via AMSTAR2.

First Author	1	2	3	4	5	6	7	8	9	10	11	12	13	14	15	16	Review Quality
Ansari 2018 [12]	Y	N	Y	PY	Y	Y	Y	PY	N/N	N	N/0	N	N	N	N	Y	Critically Low
Asgary 2014 [13]	Y	Y	Y	PY	Y	Y	N	Y	PY/0	N	Y/0	Y	Y	Y	Y	N	Moderate
Barcelos 2011 [14]	Y	PY	Y	PY	Y	Y	PY	PY	PY/0	N	0/0	0	Y	N	0	N	Moderate
Bossù 2020 [15]	Y	PY	Y	N	Y	Y	Y	PY	PY/PY	N	0/0	0	Y	Y	0	Y	Low
Chandran 2020 [16]	Y	PY	Y	PY	Y	Y	Y	PY	PY/0	N	Y/0	Y	Y	Y	Y	Y	High
Chugh 2020 [17]	Y	PY	Y	PY	Y	Y	Y	PY	PY/0	N	Y/0	Y	Y	Y	Y	Y	High
Coll 2017 [4]	Y	PY	Y	PY	Y	Y	PY	PY	PY/0	N	Y/0	Y	Y	Y	Y	Y	High
Coll 2020 [1]	Y	PY	Y	PY	Y	Y	PY	PY	PY/PY	Y	Y/Y	Y	Y	Y	Y	N	High
De Coster 2013 [27]	Y	PY	N	PY	Y	Y	PY	PY	N/N	N	0/0	0	N	Y	0	N	Critically Low
Duarte 2020 [9]	Y	PY	N	PY	Y	Y	Y	PY	PY/0	N	Y/0	Y	Y	Y	Y	Y	Moderate
Gadallah 2018 [2]	Y	PY	N	PY	Y	Y	PY	PY	PY/0	N	Y/0	Y	Y	Y	Y	Y	Low
Garrocho Rangel 2019 [18]	Y	PY	N	PY	Y	Y	PY	N	PY/0	N	0/0	0	Y	N	0	Y	Moderate
Ghajari 2008 [26]	Y	PY	Y	PY	Y	Y	N	N	PY/0	N	Y/0	Y	Y	Y	Y	Y	Moderate
Junior 2018 [11]	Y	Y	Y	PY	Y	Y	PY	PY	PY/0	N	Y/0	Y	Y	Y	Y	Y	High
Lin 2014 [16]	Y	N	Y	PY	Y	Y	PY	N	PY/0	N	Y/0	Y	Y	Y	Y	Y	Moderate
Manchanda 2020 [19]	Y	PY	Y	PY	Y	Y	Y	PY	PY/0	N	N/0	Y	Y	N	Y	Y	Low
Marghalani 2014 [29]	Y	PY	Y	PY	Y	Y	N	PY	PY/0	N	Y/0	Y	Y	Y	N	Y	Moderate
Nagendrababu 2018 [20]	Y	PY	Y	PY	Y	Y	Y	PY	PY/0	N	Y/0	Y	Y	Y	Y	N	Moderate
Najjar 2019 [30]	Y	PY	Y	PY	Y	Y	Y	N	PY/PY	N	Y/Y	Y	Y	Y	Y	Y	Moderate
Nematollahi 2019 [31]	Y	PY	N	PY	Y	Y	N	N	PY/N	N	Y/Y	Y	Y	Y	N	Y	Low
Nuvvula 2018 [21]	Y	PY	Y	PY	Y	Y	N	PY	N/0	N	0/0	0	N	N	0	Y	Low
Peng 2007 [10]	N	PY	Y	PY	Y	Y	N	PY	PY/PY	N	Y/Y	Y	Y	Y	N	N	Moderate
Pintor 2016 [32]	Y	PY	Y	PY	Y	Y	N	Y	Y/0	N	0/0	0	Y	N	0	N	Moderate
Pozos-Guillen 2016 [33]	Y	Y	Y	Y	Y	Y	N	Y	Y/0	N	Y/0	Y	Y	Y	Y	Y	Moderate
Da Rosa 2019 [22]	N	PY	Y	PY	Y	Y	N	PY	Y/0	N	Y/0	Y	Y	Y	Y	Y	Moderate
Schwendicke 2016 [34]	N	Y	N	Y	Y	Y	N	Y	Y/Y	N	Y/Y	Y	Y	Y	Y	Y	Moderate
Shafaee 2019 [35]	N	N	Y	N	Y	Y	N	PY	Y/0	N	Y/0	N	Y	Y	Y	Y	Low
Shirvani 2014 [23]	N	PY	Y	PY	Y	Y	N	Y	PY/0	N	Y/0	N	Y	N	Y	Y	Moderate
Shirvani 2014 (2) [24]	Y	Y	Y	PY	Y	Y	N	PY	Y/0	N	Y/0	Y	Y	Y	Y	Y	Moderate
Smaïl-Faugeron 2016 [6]	N	N	Y	N	Y	Y	N	N	PY/PY	N	0/0	0	Y	N	0	N	Low
Smaïl-Faugeron 2018 [3]	Y	Y	Y	Y	Y	Y	Y	Y	Y/0	Y	Y/0	Y	Y	Y	Y	Y	High
Subramanyam 2017 [25]	Y	Y	Y	PY	N	N	N	Y	Y/0	N	0/0	0	N	N	0	N	Low
Tedesco 2021 [5]	Y	Y	Y	Y	Y	Y	N	Y	PY/0	N	Y/0	Y	Y	Y	Y	Y	Moderate

0—No meta-analysis conducted, N—No, Y—Yes, PY—Partial Yes. 1. Research questions and inclusion criteria? 2. Review methods established a priori? 3. Explanation of their selection literature search strategy? 4. Did the review authors use a comprehensive literature search strategy? 5. Study selection performed in duplicate? 6. Data selection performed in duplicate? 7. List of excluded studies and exclusions justified? 8. Description of the included studies in adequate detail? 9. Satisfactory technique for assessing the risk of bias (RoB)? 10. Report on the sources of funding for the studies included in the review? 11. If meta-analysis was performed, did the review authors use appropriate methods for statistical combination of results? 12. If meta-analysis was performed, did the review authors assess the potential impact of RoB? 13. RoB accounted when interpreting/discussing the results of the review? 14. Did the review authors provide a satisfactory explanation for, and discussion of, any heterogeneity observed in the results of the review? 15. If they performed quantitative synthesis, was publication bias performed? 16. Did the review authors report any potential sources of conflict of interest, including funding sources?.

## Data Availability

Data is available in the manuscript. Any further information is available upon request on the corresponding author.

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
