# Peer review of "Vital and Nonvital Pulp Therapy in Primary Dentition: An Umbrella Review"

_jcm, 2021, doi:10.3390/jcm11010085_

Round 1
Reviewer 1 Report
There are some minor spelling and language editing mistakes, please correct them. For example: "
since there is not sufficient support to 270 recommend it [4,5,33,34,36], duo its great occurrence of complications" probably it shall be DUE after the reference, right?
Also I miss the mentioning of revascularization therapies. These are mentioned in the beginning at the materials and methods, but later no real description. If no data is available for these for primary teeth, that also needs to be mentioned.
Author Response
To the Editorial Board of
Journal of Clinical Medicine
We are pleased with the opportunity to revise and resubmit our revised manuscript titled “Vital and nonvital pulp therapy in the primary dentition: an umbrella review” (Manuscript ID jcm-1482047).
We have considered all editorial and reviewers’ comments and incorporated changes in the new revised version of the manuscript. Please find enclosed a track-changes draft of the manuscript and in addition a point-by-point rebuttal to all comments raised as outlined below. We hope that you find our responses satisfactory in addressing the criticisms and suggestions.
We hope the revised manuscript may be found acceptable for your journal.
Reviewer 1
There are some minor spelling and language editing mistakes, please correct them. For example: "since there is not sufficient support to 270 recommend it [4,5,33,34,36], duo its great occurrence of complications" probably it shall be DUE after the reference, right?
Our answer: We appreciate this remark. We have corrected this typo by replacing it with “due to” instead of “duo” (line 271).
Also I miss the mentioning of revascularization therapies. These are mentioned in the beginning at the materials and methods, but later no real description. If no data is available for these for primary teeth, that also needs to be mentioned.
Our answer: First of all, thank you for pointing out this. Unfortunately, there was a typo with the keywords that formed the search syntax. We have immediately corrected to: “(Primary teeth [MeSH] OR Pulp therapy [MeSH] OR Tooth [MeSH]) AND (Pulpotomy OR Pulpectomy OR Vital pulp therapy OR Deciduous teeth) AND (Systematic Review OR Meta-analysis)” (Lines 79-82).
Reviewer 2 Report
The text is interesting and important for pediatric dentistry.
Please check out some terms related to manuscript related like on: (qualiti of life deterioration; Caries lesions can jeopardize; unpleasant consequences; pulp is still recuperable .......), et cetera.
Author Response
To the Editorial Board of
Journal of Clinical Medicine
We are pleased with the opportunity to revise and resubmit our revised manuscript titled “Vital and nonvital pulp therapy in the primary dentition: an umbrella review” (Manuscript ID jcm-1482047).
We have considered all editorial and reviewers’ comments and incorporated changes in the new revised version of the manuscript. Please find enclosed a track-changes draft of the manuscript and in addition a point-by-point rebuttal to all comments raised as outlined below. We hope that you find our responses satisfactory in addressing the criticisms and suggestions.
We hope the revised manuscript may be found acceptable for your journal.
Reviewer 2
The text is interesting and important for pediatric dentistry.
Please check out some terms related to manuscript related like on: (qualiti of life deterioration; Caries lesions can jeopardize; unpleasant consequences; pulp is still recuperable .......), et cetera.
Our answer: We appreciate your commentary regarding these typos. We have double-checked these terms and others, and corrected them accordingly.
Reviewer 3 Report
This is a manuscript with truly relevant topic. I found this manuscript methodologically well written and informative, aimed to widen the knowledge on pulp treatment in primary dentition and highlight the available evidence-based information. Minor revisions are needed in order to improve quality of the manuscript.
Page 3, Lines 107-109: the authors stated that 59 potentially eligible full texts were screened, then 13 studies were excluded after full text screening and finally 35 reviews fulfilled inclusion criteria. However, in the Figure 1. there is n=60 screened records, 15 excluded and the final count is 33. I would kindly suggest to the authors to correct this issue regarding numbers, I believe it’s an error in the text.
Page 3, Line 116: I do not understand the sentence “All SRs covered a defined timeframe; however, one did not mention such information (9)” and I would kindly ask the authors to explain it. Is it a time frame set by the authors of this umbrella review or the time frame set by the author of the review cited under number 9 – Duarte et al, 2020. Also, when I was looking at Table 1, I did not notice anything regarding search period of this reference. maybe I did not understand well, and this is why this is important to explain, since nothing should be left to a reader to anticipate – the text should be clear and understandable.
Page 3, Line 117: The authors stated, “Three systematics reviews failed to report this language restriction [2,10,11]”, please provide explanation what language restriction?
Page 4-6, Table 1 – please provide the explanation under the table for the abbreviations NI and NR. Also, it would be helpful if the authors would cite the references in the Table 1, by adding the numbers in brackets, it would be easier for the reader to find them in the reference list.
Overall statement
The topic of the manuscript is very significant contributing to the field of dentistry, especially pediatric dentistry. In my opinion, the manuscript is methodologically very well written, with clear, well described and sound method. I believe that it will be widely cited since it could be used as a guide for systematic reviews thanks to clear and detailed description of the method protocol.
Author Response
To the Editorial Board of
Journal of Clinical Medicine
We are pleased with the opportunity to revise and resubmit our revised manuscript titled “Vital and nonvital pulp therapy in the primary dentition: an umbrella review” (Manuscript ID jcm-1482047).
We have considered all editorial and reviewers’ comments and incorporated changes in the new revised version of the manuscript. Please find enclosed a track-changes draft of the manuscript and in addition a point-by-point rebuttal to all comments raised as outlined below. We hope that you find our responses satisfactory in addressing the criticisms and suggestions.
We hope the revised manuscript may be found acceptable for your journal.
Reviewer 3
This is a manuscript with a truly relevant topic. I found this manuscript methodologically well written and informative, aimed to widen the knowledge on pulp treatment in primary dentition and highlight the available evidence-based information. Minor revisions are needed in order to improve quality of the manuscript.
Page 3, Lines 107-109: the authors stated that 59 potentially eligible full texts were screened, then 13 studies were excluded after full text screening and finally 35 reviews fulfilled inclusion criteria. However, in the Figure 1. there is n=60 screened records, 15 excluded and the final count is 33. I would kindly suggest to the authors to correct this issue regarding numbers, I believe it’s an error in the text.
Our answer: Thank you for your remark. Indeed, there is an error in the text. We replaced 59 with 60 (Line 107).
Page 3, Line 116: I do not understand the sentence “All SRs covered a defined timeframe; however, one did not mention such information (9)” and I would kindly ask the authors to explain it. Is it a time frame set by the authors of this umbrella review or the time frame set by the author of the review cited under number 9 – Duarte et al, 2020. Also, when I was looking at Table 1, I did not notice anything regarding the search period of this reference. maybe I did not understand well, and this is why this is important to explain, since nothing should be left to a reader to anticipate – the text should be clear and understandable.
Our answer: Thank you for pointing it. The reference is misidentified. The correct reference that did not mention the timeframe information was Pozos-Guillen et al. (33). It relates to the timeframe set by the authors. Where it reads “... did not mention such information (9)” it should state “... did not mention such information (33)” (Line 117).
Page 3, Line 117: The authors stated, “Three systematics reviews failed to report this language restriction [2,10,11]”, please provide explanation what language restriction?
Our answer: We removed the word “this” and substituted with “a”. The language restrictions were not set by the authors (Line 117).
Page 4-6, Table 1 – please provide the explanation under the table for the abbreviations NI and NR. Also, it would be helpful if the authors would cite the references in the Table 1, by adding the numbers in brackets, it would be easier for the reader to find them in the reference list.
Our answer: We added an explanation under the table as suggested (Lines 122 - 123). Furthermore, we were very pleased with your suggestion of adding the references. This was taken in consideration and implemented in Tables 1 and 2.
Overall statement
The topic of the manuscript is very significant contributing to the field of dentistry, especially pediatric dentistry. In my opinion, the manuscript is methodologically very well written, with clear, well described and sound method. I believe that it will be widely cited since it could be used as a guide for systematic reviews thanks to clear and detailed description of the method protocol.
Our answer: Nothing to add.